# The Expression Levels of SARS-CoV-2 Infection-Mediating Molecules Promoted by Interferon-γ and Tumor Necrosis Factor-α Are Downregulated by Hydrogen Sulfide

**DOI:** 10.3390/ijms232113624

**Published:** 2022-11-07

**Authors:** Xue Zhao, Yedi Cao, Enmin Zhao, Tiancheng Li, Tiechuan Cong, Ying Gao, Junqing Zhang

**Affiliations:** 1Department of Endocrinology, Peking University First Hospital, Beijing 100034, China; 2Department of Otolaryngology, Peking University First Hospital, Beijing 100034, China

**Keywords:** COVID-19, SARS-CoV-2, autoimmune thyroid diseases, hydrogen sulfide, cytokines

## Abstract

Autoimmune thyroid diseases (AITDs), which include Hashimoto’s thyroiditis (HT) and Graves’ disease (GD), have a higher prevalence of severe acute respiratory syndrome coronavirus 2 (SARS-CoV-2) infection in the literature. The effects of AITD-associated cytokines on SARS-CoV-2 infection-mediating molecule levels might be involved in the pathogenesis of susceptibility. We speculated that hydrogen sulfide (H_2_S) might attenuate this process since H_2_S has antiviral effects. Using immunohistochemistry, we found that angiotensin-converting enzyme-II (ACE2) expression was higher in the HT group and neuropilin 1 (NRP1) expression was higher in HT and GD groups than in the normal group, while transmembrane protease serine type 2 (TMPRSS2) expression was lower in HT and GD groups. When culturing primary thyrocytes with cytokines or sodium hydrosulfide (NaHS) plus cytokines, we found that *ACE2* and *NRP1* mRNA levels were upregulated while *TMPRSS2* levels were downregulated by interferon-γ (IFN-γ) and tumor necrosis factor-α (TNF-α). After pretreatment with NaHS in thyrocytes, ACE2 and NRP1 expression were downregulated compared to IFN-γ or TNF-α treatment, and NaHS had no effect on TMPRSS2 expression. Our findings suggested that IFN-γ and TNF-α, which are elevated in AITDs, promoted ACE2 and NRP1 expression and inhibited TMPRSS2 expression. H_2_S might protect against SARS-CoV-2 infection by downregulating ACE2 and NRP1 levels.

## 1. Introduction

The ongoing coronavirus disease 2019 (COVID-19) outbreak, which is caused by severe acute respiratory syndrome coronavirus 2 (SARS-CoV-2), continues to be a serious threat to the worldwide public health system. Angiotensin-converting enzyme 2 (ACE2) is a known cellular receptor that mediates SARS-CoV-2 entry and infection [1]. In addition to ACE2, other molecules, such as neuropilin 1 (NRP1) [2,3], serve as coreceptors that enhance SARS-CoV-2 cell entry and infectivity. Transmembrane serine protease 2 (TMPRSS2) has also been shown to enhance SARS-CoV-2 replication [4]. Multiple organ and system dysfunctions are observed during SARS-CoV-2 attacks, such as in the respiratory tract [5,6], liver [7], kidney [8], cardiovascular system [9,10], and gastrointestinal system [11]. Several potential antiviral agents might be developed into effective treatments for SARS-CoV-2 infection [12]. However, there are few effective antiviral drugs for treating or curing SARS-CoV-2 infection, and valid therapies to interfere with or combat this new virus are urgently needed [13].

It has been reported that there is a higher prevalence of SARS-CoV-2 infection in patients with autoimmune thyroid diseases (AITDs) [14]. Elevated thyroid autoantibodies are one of the most important characteristics of AITDs, which include Hashimoto’s thyroiditis (HT) and Graves’ disease (GD). Thyroglobulin antibody (TgAb) or thyroid peroxidase antibody (TPOAb) levels are elevated in serum from almost all HT patients and approximately 70% of GD patients [15,16], and TSH receptor antibody (TRAb) plays an important role in the pathogenesis of GD [17]. The reason that patients with AITDs are susceptible to SARS-CoV-2 remains unclear. In the literature, increased levels of interferon-γ (IFN-γ) [18] and tumor necrosis factor-α (TNF-α) [19] have been reported in HT, and elevated levels of interleukin-4 (IL-4) [20] and interleukin-6 (IL-6) [21] also have been shown in GD. Coperchini et al. found that IFN-γ promoted the mRNA expression of *ACE2* in human primary thyroid cells [22], and vitamin D combined with IFN-γ also increased *ACE2* expression [23]. Thus, we speculated that elevated cytokines in the thyroid of AITD patients might be a risk factor for SARS-CoV-2 infection.

Hydrogen sulfide (H_2_S), a well-known endogenous gasotransmitter, has emerged as an essential modulator of various biological functions [24,25]. Notably, it was reported that the concomitant application of N-acetylcysteine, a potential H_2_S-releasing donor, ameliorated the symptoms of COVID-19 patients [26]. Some recent reviews hypothesized that H_2_S protects against SARS-CoV-2 infection, and the specific mechanism may involve the effect of H_2_S on the expression levels of ACE2 and NRP1 [13,27,28]. Furthermore, it was reported that H_2_S suppressed TMPRSS2 expression in human airway epithelial cells, which indicated the protective effect of H_2_S against SARS-CoV-2 infection [29]. Whether H_2_S exerts similar anti-SARS-CoV-2 effects on the thyroid remains unknown. Thus, the aim of this study was to examine the effect of IFN-γ, TNF-α, IL-4, and IL-6 on the expression of SARS-CoV-2 infection-mediating molecules in the human thyroid and the potential regulatory effect of H_2_S.

## 2. Results

### 2.1. ACE2, NRP1 and TMPRSS2 Were Expressed in Human Thyroid Tissues

Immunohistochemical analysis showed that ACE2, NRP1, and TMPRSS2 were expressed in normal human thyroid cells (Figure 1A–C). The protein expression of ACE2, NRP1, and TMPRSS2 was also detected in normal human thyroid tissues (Figure 1D). Furthermore, we examined the mRNA expression of *ACE2*, *NRP1* and *TMPRSS2* in normal human primary thyrocytes and observed that NRP1 mRNA levels were higher than those of ACE2 and TMPRSS2 (Figure 1E). These results confirmed that SARS-CoV-2 infection-mediating molecules were expressed in primary normal human thyrocytes.

As the Nthy-ori 3-1-cell line is commonly used in experiments as a substitute for normal human thyrocytes, we also detected ACE2, NRP1, and TMPRSS2 expression in the Nthy-ori 3-1-cell line. We found that *ACE2*, *NRP1* and *TMPRSS2* mRNA levels were significantly higher in normal human primary thyrocytes than those in the Nthy-ori 3-1-cell line, and ACE2 and TMPRSS2 were hardly expressed in the Nthy-ori 3-1-cell line (Figure 1E), which indicated that the cell line might partially lose some thyroid phenotypes. Therefore, normal human primary thyrocytes were used for all the subsequent experiments.

### 2.2. The Expression of ACE2 and NRP1 Was Higher, and TMPRSS2 Expression Was Lower in Thyroid Tissues from AITD Patients

We further collected thyroid tissues from patients with HT, GD, and normal thyroid tissues to compare the expression levels of SARS-CoV-2 infection-mediating molecules in the thyroid. The clinical features of these enrolled subjects are shown in Table 1. The age of the GD group was significantly lower than that of the normal thyroid and HT groups. Total triiodothyronine (TT3) levels were significantly elevated in the GD group, and thyroid-stimulating hormone (TSH) levels were significantly higher in the HT group and lower in the GD group. TgAb and TPOAb levels were significantly higher in the HT and GD groups, and TRAb levels also increased significantly in the GD group compared to those in the normal thyroid group. These alterations among the three groups were consistent with the course of HT or GD disease [15,30,31].

We further compared ACE2, NRP1, and TMPRSS2 expression in thyroid tissues by immunohistochemistry (IHC). We found that thyroid tissues in the HT group exhibited significantly higher ACE2 H-scores than those in the normal thyroid group (Figure 2A). The ACE2 H-scores of the thyroid tissues in the GD group were not different from those in the normal thyroid group. (Figure 2A). In addition, we found that the NRP1 H-scores were significantly higher in the thyroid tissues from the HT and GD groups than in those from the normal thyroid group (Figure 2B). Moreover, the TMPRSS2 H-scores of the thyroid tissues in the HT and GD groups were significantly lower than those in the normal thyroid group (Figure 2C). These results indicated that the expression levels of ACE2, NRP1, and TMPRSS2 might be associated with AITDs. We speculated that the elevated expression levels of ACE2 and NRP1 in the thyroid might play an important role in SARS-CoV-2 susceptibility.

### 2.3. IFN-γ and TNF-α Increased ACE2 and NRP1 mRNA Levels and Decreased TMPRSS2 mRNA Levels

In order to verify whether the typical cytokines associated with AITDs altered the levels of ACE2, NRP1, and TMPRSS2 in the thyroid, different concentrations of IFN-γ, TNF-α, IL-4, and IL-6 were added to stimulate normal human primary thyrocytes in a concentration-dependent manner. We found that the mRNA levels of *ACE2* and *NRP1* were upregulated when thyrocytes were treated with IFN-γ or TNF-α in a dose-dependent manner (Figure 3A,B), while the mRNA level of *TMPRSS2* was decreased in thyrocytes incubated with IFN-γ or TNF-α (Figure 3C). IL-4 or IL-6 had no impact on *ACE2*, *NRP1*, or *TMPRSS2* mRNA levels (Figure 3A–C). These results demonstrated that different cytokines had different effects on the expression of SARS-CoV-2 infection-mediating molecules.

### 2.4. H_2_S Downregulated ACE2 and NRP1 Expression That Was Increased by IFN-γ and TNF-α, and H_2_S Had No Effect on TMPRSS2 Expression

In order to further evaluate the protective effect of H_2_S against SARS-CoV-2 infection-mediating molecules, normal human primary thyrocytes were divided into the control group, IFN-γ or TNF-α group, and IFN-γ or TNF-α plus H_2_S-releasing donor (sodium hydrosulfide, NaHS) group. After stimulation with IFN-γ or TNF-α, ACE2 and NRP1 protein levels were upregulated, and pretreatment of thyrocytes with NaHS reversed the effects of IFN-γ (Figure 4A) or TNF-α (Figure 4B). However, after incubation with IFN-γ or TNF-α, TMPRSS2 protein levels were downregulated in thyrocytes, and pretreating thyrocytes with NaHS had no effect on TMPRSS2 protein expression (Figure 4A,B).

## 3. Discussion

COVID-19 is a severe worldwide pandemic caused by SARS-CoV-2. This pandemic has severely stressed global socioeconomic balances and the healthcare system. To date, there have been few clinically verified antiviral drugs that are suitable for treating or curing SARS-CoV-2 infection [32,33]. SARS-CoV-2 infection requires ACE2 [1] as well as other molecules, such as NRP1 [2] and TMPRSS2 [4]. Inhibiting or attenuating the binding of the virus with these molecules is a critical step in the prevention or treatment of COVID-19.

In our study, we found that ACE2, NRP1, and TMPRSS2 were all expressed in thyroid tissues. Coperchini et al. suggested that ACE2 was localized on the cell membrane of thyroid cells [23]. We found that ACE2 expression was more abundant in the HT group, and NRP1 expression was more abundant in the HT and GD groups than that in the normal thyroid group. These results suggested that the increased levels of ACE2 and NRP1 in patients with AITDs might promote SARS-CoV-2 susceptibility in the thyroid. In our study, TMPRSS2 levels were significantly lower in thyroid tissues from AITDs than in normal thyroid tissues. In asthma nasal epithelial cells, TMPRSS2 expression was increased, and ACE2 expression was decreased compared to that in the normal group [34]. Comparing the opposite results of ACE2 and TMPRSS2 levels mentioned previously revealed that the expression levels of SARS-CoV-2 infection-mediating molecules might be tissue specific.

It was reported that patients with AITDs are more susceptible to SARS-CoV-2 infection [14]. We hypothesized that the increased cytokines in AITDs might contribute to the changes in ACE2, NRP1, and TMPRSS2 levels in the thyroid and then improve the susceptibility of SARS-CoV-2. In the current study, we found that only IFN-γ and TNF-α but not IL-4 or IL-6 promoted ACE2 and NRP1 expression. These results revealed that the characteristic cytokines in HT increased SARS-CoV-2 infection-mediating molecules, which might promote the risk of viral infection.

In the current research, we also found that ACE2 and NRP1 levels were higher, and TMPRSS2 levels were lower in the GD group than in the normal thyroid group. However, we found that the characteristic GD cytokines IL-4 and IL-6 had no effect on these molecules. In some studies, serum IFN-γ was a notable cytokine in full-blown GD [35], and TNF-α was significantly increased in thyroid tissues with GD [36,37]. Therefore, IFN-γ and TNF-α also play important roles in GD progression, and they might account for alterations in SARS-CoV-2 infection-mediating molecules in thyroid sections in the GD group.

Moreover, it was revealed that there was a cytokine storm, including interleukin-1 (IL-1), IFN-γ, TNF-α, IL-6, and others, in patients with COVID-19 [38,39]. Thus, we supposed that the abnormal inflammatory cytokines in HT increased the susceptibility of the thyroid to SARS-CoV-2, and then the cytokine storm exacerbated immune responses in patients with HT after SARS-CoV-2 infection. Subsequently, the elevated characteristic cytokines in HT further increased ACE2 and NRP1 expression in the thyroid, which formed a positive feedback loop. Conversely, we also found that IFN-γ and TNF-α attenuated *TMPRSS2* mRNA expression in thyrocytes. This result suggested that different cytokines might exert protective or pathogenic effects on SARS-CoV-2 infection-mediating molecules.

It was discovered that H_2_S had multiple pathophysiological effects and protected against damage in various organs [40,41]. Moreover, H_2_S has also been reported to suppress the replication of different viruses, such as a respiratory syncytial virus (RSV) and Nipah virus [42]. Increased H_2_S levels were reported in COVID-19 survivors [43], which suggested that H_2_S might have a beneficial effect on COVID-19. In our study, we found that H_2_S downregulated ACE2 and NRP1 expression, which was increased by IFN-γ and TNF-α. We also found that H_2_S had no effect on TMPRSS2 levels, which were decreased by IFN-γ and TNF-α. It was reported that the spike (S) glycoprotein of SARS-CoV-2 mediates viral entry by binding to ACE2, followed by proteolytic processing by TMPRSS2 and the generation of S1 polypeptides, which bind to NRP1 [2,44]. This process suggested that ACE2 served as a portal for SARS-CoV-2 entry [14,45]. Furthermore, we found that the effect of H_2_S was protective against SARS-CoV-2 infection by reducing ACE2 and NRP1. In the literature, it was reported that H_2_S attenuated ACE2 expression in rat kidneys [46] and inhibited ACE2 activity in lung tissues [47], which was consistent with our results on the H_2_S effect on ACE2.

However, some studies suggested that the downregulation of ACE2 leads to a more severe stage of COVID-19 and diabetes, and hypertensive patients with reduced ACE2 expression had higher mortality rates due to the increased activity of Ang II [48,49]. We speculated an alternative scenario. First, H_2_S treatment could alleviate hypertension [50,51], diabetes [52,53,54], atherosclerosis [55,56], etc. Second, H_2_S could reduce the effects of Ang II directly [57]. In summary, H_2_S application might alleviate SARS-CoV-2 infection by reducing ACE2 expression and decreasing the side effects of ACE2 downregulation.

There were some limitations in our current study. First, H_2_S participates in various physiological and pathological processes via multiple mechanisms [40,41,58]. We found that ACE2 and NRP1 expression was downregulated by H_2_S, but the exact mechanism remains unclear. Second, Coperchini et al. showed that the increased expression of ACE2 was on thyrocyte membranes treated with IFN-γ [23]. Due to limited thyroid tissues from humans, we could not conduct more experiments to demonstrate the localization and alteration of ACE2 on thyrocyte membranes. Moreover, we did not conduct animal experiments to verify our conclusions. More studies are needed to explain the current phenomenon.

In conclusion, we found that ACE2 and NRP1 expression was increased, and TMPRSS2 expression was reduced in thyroid tissues from patients with AITDs. IFN-γ and TNF-α promoted ACE2 and NRP1 levels and inhibited TMPRSS2 levels in human primary thyrocytes. H_2_S might protect the human thyroid against damage during COVID-19 infection by downregulating ACE2 and NRP1 levels. H_2_S-based therapeutics have advanced to the preclinical stage and might be used to alleviate cytokine-induced SARS-CoV-2 susceptibility in the thyroid.

## 4. Materials and Methods

### 4.1. Human Sample Collection

Normal human primary thyrocyte cultures were prepared using normal human thyroid tissues (*n* = 3) far from the papillary thyroid cancer (PTC) lesion that did not exhibit other thyroid diseases. Normal human thyroid tissues were also used for Western blot.

Pathological sections of human thyroid tissues were obtained from patients who underwent surgical resection due to PTC at Peking University First Hospital. Samples were collected from thyroid tissues remote from the PTC lesions. The thyroid tissue samples were confirmed as normal thyroid (*n* = 18), HT (*n* = 18), or GD (*n* = 18) by histopathological examination. None of the patients were diagnosed with COVID-19.

### 4.2. Normal Human Primary Thyrocytes and Normal Thyroid Cell Line Culture

Primary thyrocyte culture was performed as previously described [59,60]. Human thyroid tissues were digested with type II collagenase (Gibco, Grand Island, NY, USA) and 0.05% trypsin (Gibco). After being filtered through a 70 μm-pore filter and centrifuged at 500× *g* for 3 min, thyroid follicles were obtained. Thyrocytes were cultured in RPMI 1640 medium containing 10% fetal bovine serum (FBS), 1% (*v*/*v*) penicillin/streptomycin, and 1% l-glutamine (all from Gibco) at 37 °C with 5% CO_2_. The human normal thyroid cell line Nthy-ori 3-1 from Shanghai Zhong Qiao Xin Zhou Biotechnology (Shanghai, China) was cultured in the same medium.

### 4.3. Biochemical Measurements of Thyroid Function and Thyroid Autoantibodies

TT3, TT4, and TSH levels in the serum of the patients in the HT, GD, or normal thyroid groups were examined by chemiluminescence immunoassays (ADVIA Centaur; Siemens Healthcare Diagnostics, Camberley, UK), and TgAb, TPOAb, and TRAb levels in the patients’ serum were analyzed by a Cobas E601 chemiluminescence analyzer (Roche, Basel, Switzerland). The detection ranges were as follows: TT4, 58.1–140.60 nmol/L; TT3, 0.92–2.79 nmol/L; TSH, 0.55–4.78 μIU/mL; TgAb, 0–115 IU/mL; TPOAb, 0–34 IU/mL; and TRAb, 0–1.75 IU/L.

### 4.4. Immunohistochemical Staining (IHC) for ACE2, NRP1, and TMPRSS2

Human thyroid tissue slides were antigen repaired in EDTA buffer for 20 min, and endogenous peroxidase was then blocked with 3% hydrogen peroxide. After blocking with 3% bovine serum albumin (BSA, Sigma–Aldrich, St. Louis, MO, USA), the slides were incubated with the following antibodies overnight: ACE2 (1:500; Proteintech, Wuhan, China), NRP1 (1:500; Proteintech, Wuhan, China), and TMPRSS2 (1:1000, Abcam, Cambridge, UK). Afterward, the slides were incubated with the respective secondary antibodies for 60 min. Finally, positive areas were detected by 3,3′-diaminobenzidine (DAB) staining (ZSGB–BIO, Beijing, China) and observed with an Olympus BX51T microscope (Tokyo, Japan). Immunostaining intensities were assessed by histological scores (H-scores) as described previously [61].

### 4.5. Normal Human Primary Thyrocyte Treatments

In order to investigate whether cytokines affected the mRNA expression of SARS-CoV-2 infection-mediating molecules, 6.25, 12.5, 25, and 50 ng/mL of IFN-γ, TNF-α, IL-4, and IL-6 (all from Peprotech, Rocky Hill, CT, United States) were added to normal human primary thyrocytes and cultured for 24 h when the cells reached 60% confluence.

In order to determine whether H_2_S alleviated the impact of the cytokines on SARS-CoV-2 infection-mediating molecules, thyrocytes were divided into three groups: the control, cytokine, and cytokine plus NaHS groups. NaHS (Sigma–Aldrich, St. Louis, MO, USA) was added as a fast H_2_S-releasing donor, which was similar to other studies [62,63,64,65,66,67]. The cytokine group was challenged with 50 ng/mL of IFN-γ or TNF-α. The cytokine plus NaHS group was incubated with 100 μM of NaHS [60], and 50 ng/mL of IFN-γ or TNF-α was added 4 h after NaHS treatment. Finally, the cells of all three groups were collected after 48 h for Western blot analysis.

### 4.6. Real-Time PCR to Measure the mRNA Levels of ACE2, NRP1, and TMPRSS2

Total RNA was isolated from thyrocytes by TRIzol reagent (Life Technologies, Carlsbad, CA, USA). Then, the RNA was reverse-transcribed into cDNA using a High Capacity cDNA Reverse Transcription Kit (Life Technologies, Carlsbad, CA, USA). The primers for *ACE2*, *NRP1*, *TMPRSS2*, and *GAPDH* are presented in Table 2. An SYBR Green Supermix Kit (Thermo Fisher, Woolston, UK) was used to determine mRNA expression. The relative expression of each gene was normalized to GAPDH expression.

### 4.7. Western Blot Analysis of ACE2, NRP1, and TMPRSS2 Protein Levels

Human primary thyrocyte or normal thyroid tissue lysates (25 μg per sample) were separated by 10% SDS—PAGE and electrotransferred onto nitrocellulose (NC) membranes. The NC membranes were incubated with the following primary antibodies at 4 °C overnight: ACE2 (1:1000; Proteintech, Wuhan, China), NRP1 (1:3000; Proteintech), TMPRSS2 (1:2000, Abcam, Cambridge, UK), and GAPDH (1:2000; TransGen Biotech, Beijing, China). Then, the membranes were incubated with secondary antibodies (1:5000; ZSGB–BIO, Beijing, China) at room temperature for 1 h. Finally, protein bands were revealed by an enhanced chemiluminescence system (Millipore, Yonezawa, Japan). The relative gray values were analyzed by ImageJ software (Version 1.52, National Institutes of Health, Bethesda, MD, USA).

### 4.8. Statistical Analysis

All experimental data analyses were performed with SPSS 25.0 (IBM, Armonk, NY, USA). Normally distributed measurement data are expressed as the mean ± standard error (SEM) or mean ± standard deviation (SD), and nonnormally distributed data are expressed as the median and interquartile range. One-way ANOVA was used to compare means among multiple groups, and independent t-tests were applied to compare means between two groups. The Kruskal–Wallis test was used to compare medians among multiple groups. *p* < 0.05 was considered statistically significant.

## Figures and Tables

**Figure 1 ijms-23-13624-f001:**
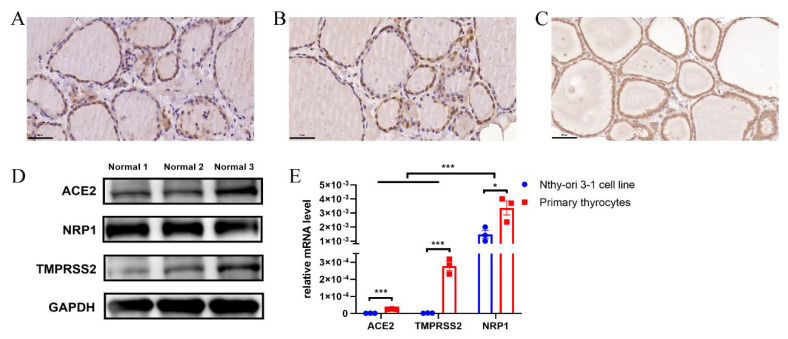
Expression of SARS-CoV-2 infection-mediating molecules in the human thyroid tissues. ACE2 (**A**), NRP1 (**B**), and TMPRSS2 (**C**) expression in human thyroid tissues was measured by immunohistochemical staining (IHC) (magnification, ×400). (**D**) ACE2, NRP1, and TMPRSS2 expression in normal human thyroid tissues were measured by Western blot. (**E**) *ACE2*, *NRP1* and *TMPRSS2* mRNA levels were measured by real-time PCR. The mRNA expression levels of *ACE2*, *NRP1*, and *TMPRSS2* were significantly higher in human primary thyrocytes than in the Nthy-ori 3-1-cell line. * *p* < 0.05, *** *p* < 0.001. ACE2: angiotensin-converting enzyme 2; NRP1: neuropilin 1; TMPRSS2: transmembrane serine protease 2; GAPDH: glyceraldehyde-3-phosphate dehydrogenase. The data are expressed as the mean ± SEM and were compared by one-way ANOVA among multiple groups or by *t*-tests between the two groups. All experiments were performed independently three times.

**Figure 2 ijms-23-13624-f002:**
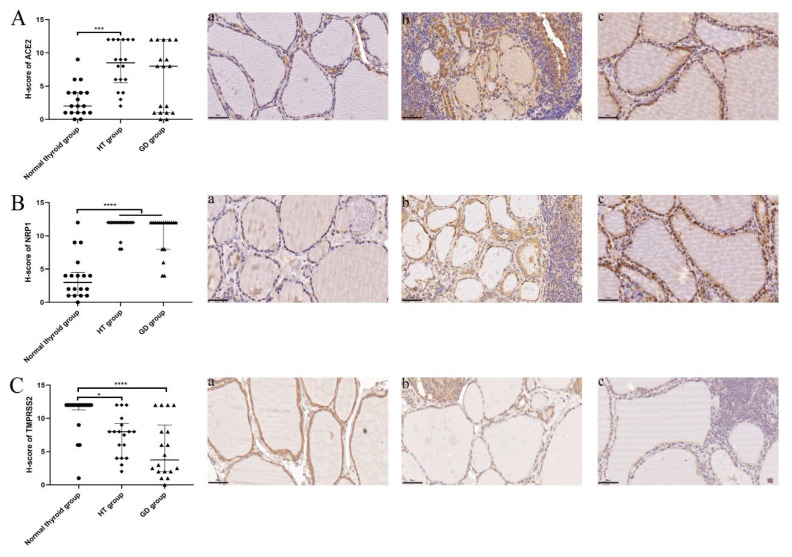
Comparison of the immunostaining scores for ACE2, NRP1, and TMPRSS2 in human thyroid tissues from the normal thyroid, HT, and GD groups. (**A**) ACE2 immunostaining scores in human thyroid tissues from the normal thyroid (a, *n* = 18), HT (b, *n* = 18), and GD (c, *n* = 18) groups were determined by IHC (magnification, ×400). ACE2 expression was significantly higher in the thyroid tissues of the HT group than in those of the normal thyroid group. (**B**) NRP1 immunostaining scores in human thyroid tissues from the normal thyroid (a, *n* = 18), HT (b, *n* = 18), and GD (c, *n* = 18) groups were determined by IHC (magnification, ×400). NRP1 expression was significantly higher in the thyroid tissues of the HT and GD groups than in those of the normal thyroid group. (**C**) TMPRSS2 immunostaining scores in human thyroid tissues from the normal thyroid (a, *n* = 18), HT (b, *n* = 18), and GD (c, *n* = 18) groups were examined by IHC (magnification, ×400). TMPRSS2 expression was significantly lower in the thyroid tissues of the HT and GD groups than in those of the normal thyroid group. * *p* < 0.05, *** *p* < 0.001, **** *p* < 0.0001. ACE2: angiotensin-converting enzyme 2; NRP1: neuropilin 1; TMPRSS2: transmembrane serine protease 2; H-score: histological score. The data are expressed as the median and interquartile range. Medians were compared by the Kruskal–Wallis test among the three groups.

**Figure 3 ijms-23-13624-f003:**
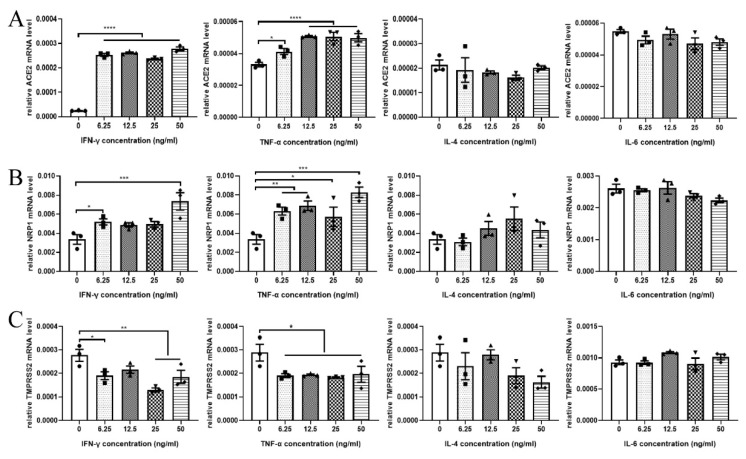
The effect of IFN-γ, TNF-α, IL-4, and IL-6 on the mRNA expression of *ACE2*, *NRP1*, and *TMPRSS2*. *ACE2* (**A**), *NRP1* (**B**), and *TMPRSS2* (**C**) mRNA expression in normal human primary thyrocytes stimulated with different concentrations of IFN-γ, TNF-α, IL-4, and IL-6 were examined by real-time polymerase chain reaction (PCR). The mRNA levels of *ACE2* and *NRP1* were upregulated in a concentration-dependent manner by IFN-γ and TNF-α in normal human primary thyrocytes, while *TMPRSS2* mRNA levels were downregulated in thyrocytes incubated with IFN-γ and TNF-α. * *p* < 0.05, ** *p* < 0.01, *** *p* < 0.001, **** *p* < 0.0001. ACE2: angiotensin-converting enzyme 2; NRP1: neuropilin 1; TMPRSS2: transmembrane serine protease 2; IFN-γ: interferon-γ; TNF-α: tumor necrosis factor-α; IL-4: interleukin-4; IL-6: interleukin-6. The data are expressed as the mean ± SEM and were compared by one-way ANOVA among multiple groups. All experiments were performed independently three times.

**Figure 4 ijms-23-13624-f004:**
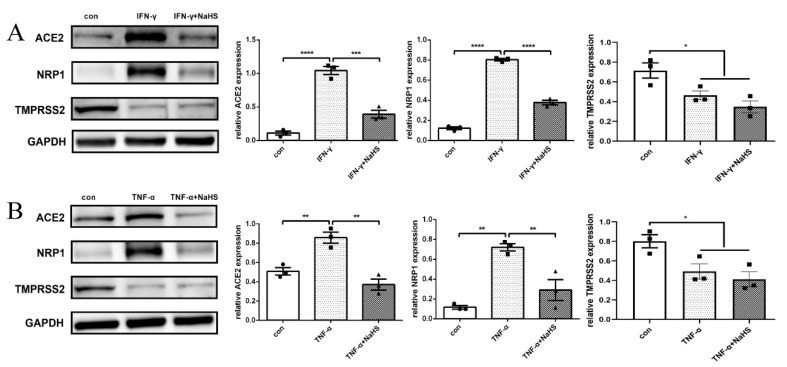
The effect of H_2_S on SARS-CoV-2 infection-promoting molecules in normal human primary thyrocytes. (**A**) ACE2, NRP1 and TMPRSS2 protein expression in normal human primary thyrocytes treated with 50 ng/mL IFN-γ or 100 μM NaHS plus IFN-γ was assessed by Western blot. ACE2 and NRP1 protein levels were upregulated by IFN-γ, and this effect was reversed by NaHS in thyrocytes. TMPRSS2 protein levels were downregulated by IFN-γ, and NaHS had no effect on TMPRSS2 protein levels. (**B**) ACE2, NRP1 and TMPRSS2 protein expression in normal human primary thyrocytes treated with 50 ng/mL TNF-α or 100 μM NaHS plus TNF-α was assessed by Western blot. ACE2 and NRP1 protein levels were upregulated by TNF-α, and this effect was reversed by NaHS in thyrocytes. TMPRSS2 protein levels were downregulated by TNF-α, and NaHS had no influence on TMPRSS2 protein levels. * *p* < 0.05, ** *p* < 0.01, *** *p* < 0.001, **** *p* < 0.0001. ACE2: angiotensin-converting enzyme 2; NRP1: neuropilin 1; TMPRSS2: transmembrane serine protease 2; GAPDH: glyceraldehyde-3-phosphate dehydrogenase; IFN-γ: interferon-γ; TNF-α: tumor necrosis factor-α; NaHS: sodium hydrosulfide. The data are expressed as the mean ± SEM and were compared by one-way ANOVA among the three groups. All experiments were performed independently three times.

**Table 1 ijms-23-13624-t001:** Demographic characteristics and clinical features of the HT, GD, and normal thyroid groups.

	Normal Thyroid Group (*n* = 18)	HT Group (*n* = 18)	GD Group (*n* = 18)
Sex (M/F)	5/13	3/15	1/17
Age (years)	43.6 ± 11.3 ^c^	44.8 ± 8.1^c^	35.2 ± 15.8 ^a,b^
TT3 (nmol/L)	1.6 ± 0.4 ^c^	1.7 ± 0.4	2.1 ± 0.7^a^
TT4 (nmol/L)	98.4 ± 21.1	99.6 ± 23.8	103.2 ± 55.1
TSH (μIU/mL)	1.7 (1.1, 2.0)	2.2 ± 1.7 ^c^	0.02 ^b^ (0.01, 1.9)
TgAb (IU/mL)	10.0 ^b,c^ (10.0, 15.2)	243.1 ^a^ (10.9, 502.2)	483.7 ^a^ (26.8, 1068.0)
TPOAb (IU/mL)	13.1 ^b,c^ (7.8, 20.5)	63.5 ^a^ (17.1, 235.6)	259.0 ^a^ (15.3, 600.0)
TRAb (IU/L)	0.7 ± 0.3 ^c^	0.8 ± 0.6 ^c^	5.04 ^a^ (2.0, 34.7)

The data are shown as the mean ± SD or median and interquartile range. Continuous variables with normal distributions were compared among the three groups using one-way ANOVA, and variables with nonnormal distributions were compared among the three groups by the Kruskal–Wallis test. The χ^2^ test was used to compare categorical variables among the three groups. HT, Hashimoto’s thyroiditis; GD, Graves’ disease; F, female; M, male; TT3, total triiodothyronine; TT4, total thyroxine; TSH, thyroid-stimulating hormone; TgAb, thyroglobulin antibodies; TPOAb, thyroid peroxidase antibodies; TRAb, TSH receptor antibodies. ^a^, *p* < 0.05 compared with the normal thyroid group; ^b^, *p* < 0.05 compared with the HT group; ^c^, *p* < 0.05 compared with the GD group.

**Table 2 ijms-23-13624-t002:** Primer sequences used for real-time PCR.

Genes		Primer Sequence (5′–3′)	Product bp
*ACE2*	Forward Reverse	GGGATCAGAGATCGGAAGAAGAAA AGGAGGTCTGAACATCATCAGTG	24 23
*NRP1*	Forward Reverse	GGCGCTTTTCGCAACGATAAA TCGCATTTTTCACTTGGGTGAT	21 22
*TMPRSS2*	Forward Reverse	GTCCCCACTGTCTACGAGGT CAGACGACGGGGTTGGAAG	20 19
*GAPDH*	Forward Reverse	GGAGCGAGATCCCTCCAAAAT GGCTGTTGTCATACTTCTCATGG	21 23

## Data Availability

Not applicable.

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
