# Peer review of "The Expression Levels of SARS-CoV-2 Infection-Mediating Molecules Promoted by Interferon-γ and Tumor Necrosis Factor-α Are Downregulated by Hydrogen Sulfide"

_ijms, 2022, doi:10.3390/ijms232113624_

Round 1

Reviewer 1 Report

This manuscript described the effect of H2S on SARS-CoV-2-related proteins in the context of autoimmune thyroid diseases (AITDs). This was a straightforward study that proposed H2S as a potential treatment for SARS-CoV-2 in AITD patients, specifically. More importantly, this study showed that the proinflammatory environment associated with AITDs increased the expression of SARS-CoV-2-related proteins and may exacerbate infection.

#1 Overall, there were some grammatical issues. Proofreading is needed before resubmission.

#2 In section 2.1, please clarify what cells were used for each experiment. Please clarify the difference between primary thyrocytes and the Nthy-ori 3-1-cell line.

#3 In section 2.1, please mention that these experiments were conducted in normal or healthy thyrocytes. Being descriptive will help readers follow along through the manuscript.

#4 Most of the detail describing each figure, including p-values, abbreviations, and statistical methods, should be included in the figure legends and not in the main text.

#5 In Table 1, the “clinical features,” including TT3, TT4, TSH, TgAb, TPOAb, and TRAb, should be discussed and explained further in the text.

#6 On lines 82-84, the use of the phrase “we investigated whether” does not make it clear that the following results are actual results. Please rewrite this sentence to clarify that these are results and not just aims of the investigation.

#7 On line 89, the phrase “might play an important role in SARS-CoV-2 infection of the thyroid” is an unfounded assumption. There is no evidence in this manuscript on SARS-CoV-2 infection.

#8 In section 2.3, please clarify that different concentrations of IL-4, IL-6, TNF-α, and IFN-γ were administered to measure ACE2 and NRP1 mRNA expression in a dose-dependent manner.

#9 Please include the full western blot images for Figure 4A and 4B.

#10 I suggest moving section 4.4 earlier in the methods (to be section 4.2) to clarify what cells were used for the subsequent analyses.

Reviewer 2 Report

Zhao et. al have precisely studied the effect of IFN-γ, TNF-α, IL-4 and IL-6 on the expression of SARS-CoV-2 infection-mediating molecules in the human thyroid and the potential regulatory effect of H2S.

 Here are my comments regarding the same.

1.       Along with the IHC analysis on thyroid tissues, western blot analysis is also needed to show the expression of ACE2, NRP1 and TMPRSS2. (Fig1)

2.       Table 2 explains the clinical features of the patients, what was the COVID 19 status of these patients during the study? If they are positive then is the IFN-γ, TNF-α, IL-4 and IL-6 high in these patients.

3.       Figure 3 shows the expression level of ACE2, NRP1 and TMPRSS2 in the patients. What happens to the protein level would also be interesting to know.

 If the author provide these results then the story would be complete to conclude the relation with COVID19

Reviewer 3 Report

This work suggests hydrogen sulfide (H2S) as a potential antiviral treatment against SARS-CoV-2 infection through its downregulating effects on the angiotensin converting enzyme-2 (ACE2) and neuropilin-1 (NRP1) expression by the thyrocytes. It fits within the scope of the journal and my suggestion to the authors is to add a discussion on the possibility (likeliness or unlikeliness) of observing the same affect of H2S on the other types of ACE2 expressing cells. Also, the authors used hydrosulfide (NaHS) to represent H2S. Accordingly, the similarities and the differences, or the (metabolic)connections in between the (effects of) H2S and NaHS should better be discussed as well.

Round 2

Reviewer 1 Report

Thank you for your response to my review. The authors have sufficiently addressed all my concerns and I recommend the manuscript for publication. 

Reviewer 2 Report

Great Story!

The authors have performed the suggested experiments and have concluding results. 

with the new experiments and results added I think the story is more precise.